# Mode Coupling at around M-Point in PZT

**DOI:** 10.3390/ma15010079

**Published:** 2021-12-23

**Authors:** Sergey Vakhrushev, Alexey Filimonov, Konstantin Petroukhno, Andrey Rudskoy, Stanislav Udovenko, Igor Leontyev, Alexei Bosak

**Affiliations:** 1The Ioffe Physical-Technical Institute of the Russian Academy of Sciences, 194021 Saint-Petersburg, Russia; 2Higher School of Engineering Physics, Peter the Great St. Petersburg Polytechnic University, 195251 Saint-Petersburg, Russia; filimonov@rphf.spbstu.ru (A.F.); k.a.petroukhno@gmail.com (K.P.); rector@spbstu.ru (A.R.); s_udovenko@mail.ru (S.U.); 3The Faculty of Physics, Southern Federal University, 344006 Rostov-on-Don, Russia; i.leontiev@rambler.ru; 4European Synchrotron Radiation Facility (ESRF), 38000 Grenoble, France; alexei.bossak@esrf.fr

**Keywords:** soft mode, perovskites, mode coupling, antiferrodistortive mode

## Abstract

The question of the microscopic origin of the M-superstructure and additional satellite peaks in the Zr-rich lead zirconate-titanate is discussed for nearly 50 years. Clear contradiction between the selection rules of the critical scattering and the superstructure was found preventing unambiguous attributing of the observed superstructure either to the rotation of the oxygen octahedra or to the antiparallel displacements of the lead cations. Detailed analysis of the satellite pattern explained it as the result of the incommensurate phase transition rather than antiphase domains. Critical dynamics is the key point for the formulated problems. Recently, the oxygen tilt soft mode in the PbZr${}_{0.976}$Ti${}_{0.024}$O${}_{3}$ (PZT2.4) was found. But this does not resolve the extinction rules contradiction. The results of the inelastic X-ray scattering study of the phonon spectra of PZT2.4 around M-point are reported. Strong coupling between the lead and oxygen modes resulting in mode anticrossing and creation of the wide flat part in the lowest phonon dispersion curves is identified. This flat part corresponds to the mixture of the displacements of the lead and oxygen ions and can be an explanation of the extinction rules contradiction. Moreover, a flat dispersion surface is a typical prerequisite for the incommensurate phase transition.

## 1. Introduction

In the last few decades, the interest in antiferroelectric (AFE) materials [1] has grown sharply. This interest is associated both with their potential practical applications [2] and with the very rich physics of these materials [3]. Probably the most widely studied are pure lead zirconate PbZrO${}_{3}$ [4] and the lead zirconate titanate solid solutions with low Ti content PbZr${}_{1-x}$Ti${}_{x}$O${}_{3}$ (PZT) [5]. Lead zirconate is a classic perovskite-like antiferroelectric, while lead titanate is perovskite-like ferroelectric. These two compounds produce continuous system of solid solutions with x from 0 to 1. The phase diagram of the PZT at x≤0.06 was reported in the papers [6,7] and it was shown that these compounds are antiferroelectrics (AFE) at room temperature. All those compounds have a simple cubic (C) structure in the paraelectric phase. In most of these materials, the phase transition to the AFE phase occurs through an intermediate ferroelectric phase (IFE) [8,9,10]. The phase transitions from the paraelectric phase to the AFE one occurs through the IFE phase [8,9,10]. A wide variety of structural states is observed in compounds of the PZO group, including the aforementioned AFE and IFE phases, as well as incommensurate structures [11,12]. The structure of the AFE phase can be considered as well established, while the structure of the intermediate FE phase remains questionable, despite of the may papers on this subject (see Ref. [10] and references there). The analysis of the structure of the IFE phase is complicated by the coexistence of the order parameters. If we assume that the transition from the paraelectric (PE) to the FE phase is intrinsic ferroelectric, then polarization will be the primary order parameter. However, it has been shown that in the FE phase superstructure reflexes occur at the M point of the Brillouin zone (BZ) $\left(h+\frac{1}{2}\;k+\frac{1}{2}\;l\right)$, and additional satellite reflexes occur near it [13,14,15]. The arising M-superstructure cannot be correlated with the antiferrodistortive (AFD) order parameter (OP) associated with parallel rotations of oxygen octahedra in neighboring planes since there is no h=k extinction law specific for such an OP [16]. Moreover, as shown in the paper [14], the emerging complex set of satellite reflections does not fit into the picture of antiphase domains proposed in [15,17], but, rather these satellites should be attributed to the “true” incommensurate phase. To understand how the structure of the FE phase is formed, one should trace the behavior of the relevant critical dynamics. Lattice dynamics of the antiferroelectric and ferroelectric materials of the PZO group were studied both experimentally [11,18,19,20,21,22,23,24,25] and theoretically [26,27,28,29,30]. In addition to the direct measurements of the phonon spectra using the optical spectroscopy and inelastic X-ray scattering (IXS) complementary diffuse scattering (DS) studies were done [14,31,32,33,34] The main attention was paid to the dynamic origin of the AFE phase and the details of the phonon dispersion around the Σ-point ($q=(\frac{1}{4}\;\frac{1}{4}\;0$) and the R-point ($q=(\frac{1}{2}\;\frac{1}{2}\;0$).

The phonon dynamics at and around the M-point were less studied and remains less understood. Hlinka et al. [19] reported the results of the inelastic X-ray study of the morphotropic PZT. Temperature-dependent inelastic scattering was found at the momentum transfer Q=(2.50.50) that was attributed to the soft $M_{5^{\prime }}$ mode (here and below [35] notations are used for the mode symmetry). Actually, this assignment is questionable since M${}_{3}$ oxygen AFD mode was not taken into consideration at all. Paściak et al. [31] reported the results of the X-ray diffuse scattering study of the pure PZO at 500 K. Strongly asymmetric distribution of the DS intensity near the momentum transfers $\vec{Q}=\left(h+\frac{1}{2}\;k+\frac{1}{2}\;l\right)$ were observed for h≠k with a maximum on one side of the M-point and a minimum on the other side. At the positions with h=k there were clear minima of the DS intensity. Similar results were obtained for the PZT with 0.7% of Ti (PZT0.7) [33]. The conclusion was made, that the observed scattering features are related to the “interactions between the Pb-related modes and the octahedra tilting modes”. In the later neutron DS experiment [32] no asymmetry in the distribution of the scattering intensity was reported. Temperature evolution of the X-ray DS intensity in the cubic phase of the PZT0.7 was traced by Andronikova et al. [14]. It was found that the peak near the M-point grows fast on approaching the transition to the FE phase. The direct IXS measurements have demonstrated the dynamic nature of the observed DS feature. In the very recent paper [25] the results of the study of the temperature evolution of the phonon spectra of the PZT2.4 are presented. The M${}_{3}$ AFD mode was clearly identified softening on cooling to the phase transition from C to IFE phase. It was shown, that the frequency of that mode shows critical behavior $\omega_{M3}=A\left(T-T_{c}^{AFD}\right)$ with critical temperature $T_{c}^{AFD}$ = 438 K. At T < 700 K the frequency of the M${}_{3}$ mode became lower, than the frequency of the acoustic. Aside from the M-point the AFD and transverse acoustic modes have the same symmetry ($\Sigma_{3}$ in the particular case of the (110) direction) and the mode coupling between these modes should arise to prevent the crossing of these phonon branches.

Ab-initio calculations show that the M${}_{3}$ mode is the most negative one [26,36], however this instability is supposed to be less important than the instabilities at the Brillouin zone center or at the R-point. In the paper [37] possible coupling of the M${}_{3}$ and M${}_{5^{\prime }}$ modes was discussed and the conclusion was made about the low probability of simultaneous condensation of both types of M-like modes M${}_{3}$ and M${}_{5^{\prime }}$ in the ferroelectric phase. It was stated that: “This finding suggests it is highly unlikely that two types of M-like modes M${}_{3}$ and M${}_{5^{\prime }}$ both condense in the ferroelectric phase, as suggested in Ref. [12]”. Instead, the experiments probably reflect the fact that M${}_{3}$ and M${}_{5^{\prime }}$-like modes mix with each other due to the breaking of cubic symmetry [37]. It is necessary to mark out that indeed in the m$\bar{3}$m cubic phase M${}_{3}$ and M${}_{5^{\prime }}$ modes cannot mix, but at arbitrary small distance from the M-point they acquire the same symmetry and the coupling becomes allowed, moreover as it is mentioned above, such coupling becomes inevitable in the case when the frequency of the M${}_{3}$ is lower than that of the M${}_{5^{\prime }}$ mode.

To clarify the question about a mode coupling in the vicinity of the M-point we performed the detailed study of the phonon spectra in PZT2.4 using the IXS technique.

## 2. Materials and Methods

The PbZr${}_{0.976}$Ti${}_{0.024}$O${}_{3}$ (PZT2.4) single crystals were grown in the Southern Federal University [7]. Crystals were grown from the solution of the PZT in the PbO-B${}_{2}$O${}_{3}$ melt. A 57 cm${}^{3}$ platinum crucible was used. Temperature gradient along the crucible height was about 10–20 K/cm with a crucible base temperature of about 1250 K. Crystallisation was completed in 75–100 h. Single crystals were washed from the cooled melt with a dilute acetic acid. Rod-shaped samples of about 50*50*300 μm${}^{3}$ were cut using the diamond saw, polished with the diamond paste and etched in the hydrochloric acid to remove the damaged surface layer.

According to the phase diagram PZT2.4 undergoes phase transition from the paraelectric cubic phase to a ferroelectric one at around 500 K and on further cooling to the AFE phase at around 423 K. Transition temperatures were checked on the sample from the same batch by following the temperature evolution of the intensity of the superstructure peaks.

Inelastic X-ray scattering measurements were performed at the ID28 spectrometer at the ESRF. The sample was mounted on a sample holder using a cement glue with [001] axis vertical, so that all the measurements were done in (hk0) plane Due to the high quality of the sample and very “clean” there was no visible quasi-elastic contribution. High temperature data not shown in the paper clearly confirm it. Si(11 11 11) monochromator with an energy resolution of 1.5 meV FWHM was used. Temperature was controlled with the special heat blower with the precision of about ±0.5 K.

## 3. Results

Most of the measurements were done in the vicinity of the $Q_{M1}$ = (3.5 0.5 0) for the wavevector transfers $Q_{1}=Q_{M1}+\left(-qq0\right)$ at 5 temperatures: T = 700, 650, 625, 575 and 525 K. Observation of the $M_{3}$ soft mode, was earlier reported at $Q_{M1}$. At each temperature Q-constant scans were performed for 7 q-values: −0.15; −0.1; −0.05; 0; 0.05; 0.1; 0.15. Additional reference data at 700 K were collected at around $Q_{M2}$ = (2.5 2.5 0) were the M${}_{3}$ mode is not observable. In Figure 1 2-d plots of the inelastic data in the coordinates q–E are presented for 625 and 550 K.

As reported earlier on approaching the transition temperature strong increase of the inelastic scattering intensity is seen aside of the M-point. Important fact is that the inelastic signal at positive and negative values of q are substantially different. Distribution of the IXS intensity around $Q_{M2}$ is obviously symmetric and there is no peculiarities aside M-point. Presented data allow to attribute reported earlier asymmetry [14,31] of the energy integrated diffuse scattering to the dynamic effects and the difference of the results near $Q_{M1}$ and $Q_{M2}$ indicates an importance of the $M_{3}$ mode. We have analyzed Q-constant scans at both sides of the M-point. In the Figure 2 the set of Q-constant scans measured at 550 K is shown. At this temperature the difference of the spectra at +q and −q is most evident.

It is clearly seen that both intensity and the shape of the spectra are different at different sides of an M-point. Obviously, the frequencies of all phonons are the same for positive and negative *q* and the only possible origin of the asymmetry of the spectra is the difference of the inelastic structure factors given by Equation (Equation 3). To explain the observed effect the model based on the coupling of the TA and oxygen octahedra rotation modes was developed.

### 3.1. Mode Coupling Model

As was explained in the paper [25] low frequency part of the IXS spectra at M-point includes 3 modes (the notation from the [35] is used):2 double degenerate M${}_{5^{\prime }}$ modes: acoustic M${}_{5^{\prime }}^{A}$ and optic M${}_{5^{\prime }}^{O}$nondegenerate M${}_{3}$ oxygen mode

M${}_{3}$ and M${}_{5^{\prime }}^{A}$ modes have close frequencies, but due to the different symmetries these modes do not interact. When moving aside of M point along Σ line M${}_{5^{\prime }}$ modes split into longitudinal $\Sigma_{1}$ and transverse $\Sigma_{3}$ modes and M${}_{3}$ phonons acquire $\Sigma_{3}$ symmetry. So transverse acoustic (TA) and “oxygen” (Actually aside from M point the mode cannot be considered as purely “oxygen mode” but in the proposed model we neglect displacements of other ions) modes now have the same symmetry and may couple. Actually, this coupling is unavoidable since on approaching the transition the M${}_{3}$ mode goes below the M${}_{5^{\prime }}^{A}$ one and aside the M point the modes should intersect. Since two modes of the same symmetry cannot have the same frequencies mode coupling should arise resulting in the mode anticrossing. We have used the standard formalism [38] to describe the inelastic scattering on the coupled transverse acoustic and oxygen modes. Scattering intensity on the two coupled modes $I_{CM}$ in this case can be written as:(1)$$I_{CM}=Sn\left(\omega \right)\sum_{i,j=1,2}F_{i}^{*}\left(\vec{Q}\right)G_{ij}F_{j}\left(\vec{Q}\right)$$

Here *S* is the scaling factor, n(ω)—Bose factor, indexes *i* and *j* correspond to the transverse acoustic (i,j=1) or oxygen (i,j=2) modes, $F_{i}$—inelastic structure factors for the corresponding modes, and the dynamic susceptibility for two coupled anharmonic oscillators is described as:(2)$$G_{ij}^{-1}=\left(\begin{array}{cc} \omega_{1}^{2}-\omega^{2}+i\Gamma_{1}\omega & \Delta_{12}+i\Gamma_{12}\omega \\ \Delta_{12}+i\Gamma_{12}\omega & \omega_{2}^{2}-\omega^{2}+i\Gamma_{2}\omega \end{array}\right)$$
$\omega_{i}$ and $\Gamma_{i}$ frequencies and dumping constants of uncoupled modes and $\Delta_{12},\;i\Gamma_{12}$ are real and imaginary parts of the complex damping constant. The presented model contains 8 parameters that is one more than can be determined from the spectra. This fact was first recognized by Barked and Hopfield [39] and discussed later [38]. Usually to overcome the uncertainty the interaction is considered either purely elastic ($\Gamma_{12}=0$) or purely viscous damping ($\Delta_{12}=0$). In our case, the $F_{i}$ for the TA, LA and AFD modes was directly calculated using the formula:(3)$$F_{i}\left(\vec{Q}\right)=\sum_{l}\frac{f_{l}\left(Q\right)}{\sqrt{M_{l}}}\left(\vec{Q}{\vec{e_{l}}}^{j}\left(\vec{q}\right)\right)\exp\left(i\vec{Q}\vec{r_{l}}\right)$$
$f_{l}\left(Q\right)$ is the atomic scattering factor of the ion *l* and the $M_{l}$ the mass of this ion. ${\vec{e_{l}}}^{j}\left(\vec{q}\right)$ is the corresponding to the ion *l* component of the eigenvector of the mode *j* at the reduced wavevector $\vec{q}$, $\vec{r_{l}}$ the position of the ion *l* inside the lattice cell. Summation is made over the displacements of the ions in the mode *i*. We have neglected here the Debye-Waller factor considering it the same for all the ions and including it into the scaling parameter *S*. It is generally accepted that in PZT the acoustic modes near the Brillouin zone boundary are dominated by the displacements of the lead ions [31]. So the eigenvectors for the TA and LA modes were taken as: ${\left({\vec{e}}_{Pb}\right)}^{TA}=\left(\frac{1}{\sqrt{2}}-\frac{1}{\sqrt{2}}0\right)$ and ${\left({\vec{e}}_{Pb}\right)}^{LA}=\left(\frac{1}{\sqrt{2}}\frac{1}{\sqrt{2}}0\right)$. We also considered, that not too far from the M-point, the $\Sigma_{3}$ mode, which is a continuation of the $M_{3}$ one, is completely determined by the oxygen displacements. The eigenvectors for the O${}_{1}$ at $r_{O1}=\left(\frac{1}{2}0\frac{1}{2}\right)$ were assumed as ${\vec{e}}_{O1}=\left(100\right)e^{i\phi }$ and for the O${}_{2}$ at $r_{O2}=\left(0\frac{1}{2}\frac{1}{2}\right)$ as ${\vec{e}}_{O2}=\left(0-10\right)e^{i\phi }$. Eigenvectors of the M${}_{3}$ mode are real and aside of it are supposed to be complex, so the phase ϕ was included. Eigenvectors at positive and negative *q*’s obey the relation ${\vec{e}}^{*}\left(\vec{q}\right)=\vec{e}(-\left(\vec{q}\right)$ [40].

### 3.2. Data Treatment

Experimental spectra were approximated by the expression similar to Equation (Equation 1) with i,j=1,...,4. As explained above 4 modes were included: (1)—TA mode with eigenvector ${\left({\vec{e}}_{Pb}\right)}^{TA}$; (2) — “oxygen” AFD mode with eigenvectors ${\vec{e}}_{O1}$ (3)—LA mode with eigenvector ${\left({\vec{e}}_{Pb}\right)}^{LA}$; and ${\vec{e}}_{O2}$ and (4)—optic mode with unknown eigenvectors. Generalized susceptibility was calculated similarly to Equation (Equation 2). Interaction was included only for the modes 1 and 2:(4)$$G_{ij}^{-1}=\left(\begin{array}{cccc} \omega_{1}^{2}-\omega^{2}+i\Gamma_{1}\omega & \Delta_{12}+i\Gamma_{12}\omega & 0 & 0\\ \Delta_{12}+i\Gamma_{12}\omega & \omega_{2}^{2}-\omega^{2}+i\Gamma_{2}\omega & 0 & 0\\ 0 & 0 & \omega_{3}^{2}-\omega^{2}+i\Gamma_{3}\omega & 0\\ 0 & 0 & 0 & \omega_{4}^{2}-\omega^{2}+i\Gamma_{4}\omega \end{array}\right)$$

For modes 1, 2 and 3 inelastic structure factors were calculated according to (Equation 3) and for the mode 4 $F_{4}$ was free independent parameter. Spectra for all temperature (5) and all q-vectors (7) were fitted simultaneously. Dispersion of the “oxygen” mode was described by the simple scaling relation:(5)$$\omega_{2}^{2}\left(q\right)=\omega_{2}^{2}\left(0\right)+Dq^{2}=\omega_{2}^{2}\left(0\right)\left(1+\frac{D}{\omega_{2}^{2}\left(0\right)}\right)=\omega_{2}^{2}\left(0\right)\left(1+r_{c}^{2}q^{2}\right)$$

Here the only temperature dependent parameter is is the frequency of the $M_{3}$ at the M-point $\omega_{2}\left(0\right)$. It was taken from the paper [25] where it was determined from the analysis of the data at M-point, where the mode coupling is forbidden. D is temperature independent coefficient and $r_{c}$ is the correlation length of the “oxygen” mode. Most of the parameters were considered temperature independent. As temperature dependent were set: (1)—scale factor *S* (*q*-independent), real and imaginary parts of the coupling constant: $\Delta_{12}$ and $\Gamma_{12}$ (both T and *q*-dependent). Python interface iminuit [41] was used to implement the variable metric method [42].

## 4. Discussion

Proposed model gives a very good description of the observed IXS spectra. The difference in the shape of the IXS spectra at two sides of the M-point is completely determined by the contribution of the interference term $F_{1}^{*}ImG_{12}^{-1}F_{2}+F_{1}ImG_{21}^{-1}F_{2}^{*}$. This agrees with our earlier assumption [43] that the asymmetry of the diffuse scattering in pure PZO and Zr-rich PZT has the dynamic nature. The most important effect of the mode coupling is the renormalization of the frequencies of the interacting modes. In Figure 3a the *q*-dependence of the frequencies of the “bare” (unrenormalized by the mode coupling) TA and AFD modes are shown. At temperatures below 700 K the dispersion curves intersect.

This intersection is unrelated to the specific details of the shape of the phonon dispersion curves since the TA branch at the zone boundary is always rather flat, while the oxygen AFD mode always demonstrates a steep frequency increase. Since TA and AFD modes have the same $\Sigma_{3}$ symmetry they cannot have the same frequency at the same wavevector so the mode interaction should renormalize their frequencies. The frequencies of the perturbed modes were calculated as the eigenvalues of the real part of matrix (Equation 2) [38]. The obtained dispersion curves are presented in Figure 3b. Mode coupling results in the mixing of the eigenvectors so we cannot any identify them as acoustic and AFD modes but will address the perturbed excitations as the Upper (U) and Lower (L) modes. Polarization vectors of the perturbed modes ${\vec{e}}^{p}$ can be calculated using the matrix of the eigenvectors of the matrix (Equation 2) $S_{ij}$ as:(6)$${\vec{e}}_{i}^{p}=\sum_{ij}S_{ij}{\vec{e}}_{j}^{0},$$
here ${\vec{e}}^{0}$—polarization vectors of the unperturbed modes defined above at the end of the Section 3.1. In Table 1 the value of $S_{11}$, which determine the contribution of the TA mode to the lower perturbed mode is shown. It should be mentioned that the eigenvectors, constituting the $S_{ij}$ are orthonormal, so the equal contributions of the TA and AFD modes to the perturbed modes correspond to the situation when absolute values of all elements in $S_{ij}$ are equal to the $\sqrt{2}$.

At 700 K AFD mode frequency at the M-point is practically equal to that of the acoustic mode. There is no mode anticrossing, but the shape of the TA branch dispersion is affected by the mode coupling, resulting in the mode repulsion. The observed “ugly” TA branch is in good agreement with the paper [43] on PZO0.7. There is a large contribution of the oxygen displacements L-mode at around q=0.05 since the frequencies of the unperturbed excitations are very close to each other. As it is seen from Table 1 already at 650 K L-mode starts as purely AFD one at M-point and gradually is transformed into the TA mode. At 550 K L-mode is purely AFD at q=0.05 and at q=0.1 the contributions of the lead and oxygen displacements are nearly equal. The L-mode at 550 K is absolutely flat in the *q*-range from −0.05 to 0.05. Such flat part of the dispersion curve is typical prerequisite for the incommensurate phase transition. Moreover mode anticrossing should exists for the deviation from the M-point $\vec{Q}=(h+0.5\pm \zeta_{1}\;k+0.5\pm \zeta_{2}\;l\pm \zeta_{3}$ in any direction, except $\zeta_{1}=\zeta_{2}=0;\;\zeta_{3}\neq 0$, corresponding to M-R line. So we may expect that weakly q-dependent mixed mode exists in a very large volume of the reciprocal space around M-point. In the paper [36] the coupling of the M${}_{3}$ and $\Gamma_{15}$ (ferroelectric soft mode) distortions were discussed. The statement was made: “...the presence of M3 distortions is secondary and incidental, and by itself does not significantly stabilize ferroelectric distortions...” [36]. Obtained results ask to reconsider this point. In reality, instead of the soft M${}_{3}$ mode at the isolated point of the Brillouin zone, there is mixed soft mode spread over large volume of the q-space. This should result in the specific low frequency peak in the phonon density of states that can be a key factor determining the structure of the intermediate phase in Zr-rich PZT.

And finally we would like to emphasize one more point that is outside the main scopes of this paper. Analyzing the contributions to the spectra from different excitation (see Figure 2) and the interference term (see Figure 2) we can see that the IXS intensity from the AFD mode is quite comparable from that from the TA mode. The contribution of the mixed interference term $F_{1}^{*}ImG_{12}^{-1}F_{2}+F_{1}ImG_{12}^{-1}F_{2}^{*}$ is even more important. These results confirm the high efficiency of the IXS technique for studying of the soft AFD modes in the perovskite-like materials as it was discussed in our previous paper [25]. In the case of the inelastic neutron scattering, the contribution of the oxygen mode will be dominating even at equal frequencies of the AFD and acoustic modes, while if the oxygen mode goes far below the acoustic one, the study of the mode coupling with neutrons becomes problematic. This question deserves separate specialized study which is planned for the future.

## 5. Conclusions

The IXS study of the low frequency dynamics of the paraelectric phase of the PZT2.4 was performed. We developed the model describing the main features of the observed spectra by the coupling of the TA and AFD oxygen modes. Proposed model provided quantitatively good agreement with the experimental data. The coupling of the TA and AFD modes fully explains the the IXS scattering intensity and lineshapes. The mode interaction results in anticrossing of the phonon branches and the creation of the mixed branch with the displacements pattern gradually evolving from the pure oxygen displacements to the predominantly lead displacements. The wide central part of this branch is completely flat. Taking into account that the mode coupling is not limited to the (qq0) direction from the M-point the conclusions can be made about the existence of the mixed flat soft branch spread over a wide q-range. Such flat mixed soft branch can be considered as the dynamic origin incommensurate order in the intermediate phase. 

## Figures and Tables

**Figure 1 materials-15-00079-f001:**
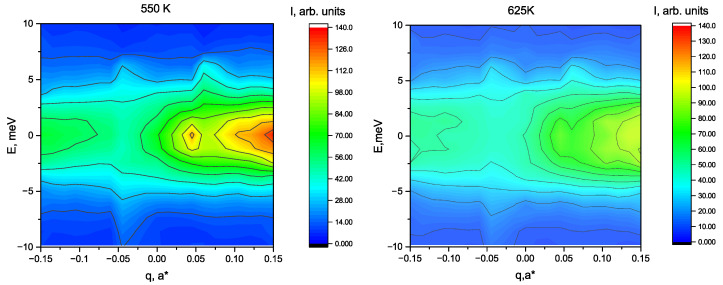
Two-dimensional plots of the X-ray inelastic spectra at 625 K and 550 K. q—is the deviation from the M-point and E—energy transfer in meV.

**Figure 2 materials-15-00079-f002:**
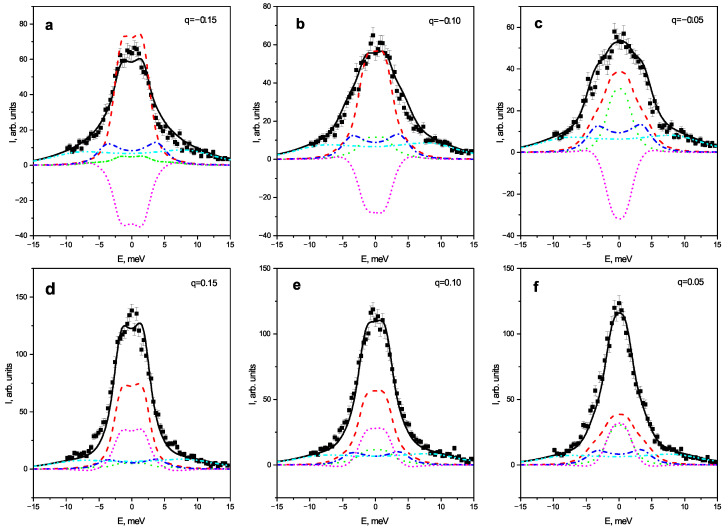
IXS spectra of PZT2.4 at two sides of M-point at 550 K. Panels (**a**–**c**) correspond to the negative q values −0.15 (**a**), −0.10 (**b**) and −0.05 (**c**), and panels (**d**–**f**) to the positive q values 0.15 (**d**), 0.10 (**e**) and 0.05 (**f**). The experimental data are represented by squares with error bars, solid lines correspond to (black in the online version) the overall fit, dashed (red) and dash-dotted (blue) lines show contributions of the unperturbed TA and LA modes, while loosely dotted lines (green) represent the AFD mode, dash dot dot (cyan) lines are used for high-energy optic mode, and dotted lines (magenta) show the contribution of mode coupling.

**Figure 3 materials-15-00079-f003:**
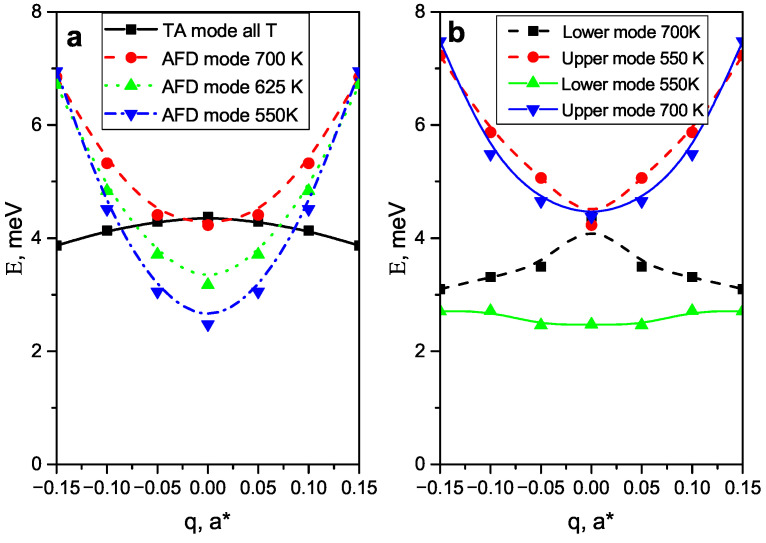
(**a**)—Dispersion curves of the “bare” TA and AFD modes. Black squares and solid black line—temperature independend TA mode; red circles dashed line—AFD mode 700 K, green triangles, dashed line—AFD mode 625 K; blue triangles dash dotted line—AFD mode 550 K. (**b**)—dispersion of the renormalized modes—color and line types are shown on the fiigure).

**Table 1 materials-15-00079-t001:** The contribution of the unperturbed TA phonon mode to the lowest (L) perturbed mode polarisation vectors (see Equation (Equation 6)). Equal contributions corresponds to the $|S_{11}|=\sqrt{2}$.

Temperature K	$|S_{11}|\;q=0.00$	$|S_{11}|\;q=0.05$	$|S_{11}|\;q=0.10$	$|S_{11}|\;q=0.15$
700	1	0.74	0.86	0.935
650	0	0.63	0.82	0.93
625	0	0.58	0.80	0.92
575	0	0.49	0.77	0.92
550	0	0.46	0.76	0.92

## Data Availability

The data presented in this study are available on request from the corresponding author.

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
