# Peer review of "Mode Coupling at around M-Point in PZT"

_materials, 2021, doi:10.3390/ma15010079_

Round 1
Reviewer 1 Report
The reviewed manuscript concerns the results obtained for PZT. I have the following critical remarks concerning this work:
- In keywords, the word “perovskites” is given, but, only one this word was mentioned in the main text. In the Introduction, an explanation about the studied compound should be added.
- In chapter 2 (lines 90 and 91) is written, “The PbZr976Ti0.024O3 (PZT2.4) single crystals were grown in the Southern Federal University [7].”. This sentence should be improved. Ref. [7] is also cited in the Introduction, but information about the studied compound isn’t mentioned. In this publication, studies are new or repeated from Ref. [7]? It should be clarified.
- In chapter 2, p. 3 (lines 93, 94), it is written, “Crystallisation was completed in 75 – 10”. I suppose that in this sentence is a mistake.
- In chapter 3 (lines 110, 111), is “T=700 K, 650 K, 625 K, 575 K and 525 K”, but should be “T=700, 650, 625, 575 and 525 K”, on p. 3 (line 115) and in the title of Figure 1 and 3 is “625 K and 550 K”, but should be “625 and 550 K”.
- On p. 6 (line 159, 160), is “Python interface iminuit [41] was used to implement variable metric method [42] from the MINUIT C++ package.” Ref. for “the MINUIT C++ package” should be added.
- In scientific papers, it doesn’t use “we” (e.g. in Abstract, lines 131, 136, 212, 213), but sentences in passive voice.
- On p. 2 (lines 79-82) and p. 8 (lines 193-104), cited sentences should be written in their own words with numbers of References given in brackets.
- Figure 2 should be improved. Some lines are not well visible. The description in the title of Figure 2 isn’t clear.
- For the first time, an abbreviation with the full name should be given in the text (in each of three sections: the abstract; the main text; the first figure or table), and next, only an abbreviation could be used, e.g. on p. 1 (line 24) is “an intermediate ferroelectric phase (IFE).” and (lines 25-26) “the intermediate ferroelectric (IFE) phase”.
- Editorial mistakes should be corrected e.g.
- in some places in the text the space between words should be added, e.g. p. 1 (line 27), “group,including”, “Ref.[10]”, p. 2 (line 38), “OP[16]”, p. 4, “and"oxygen"”, p. 5 (line 144) “ions[31]”, p. 6 (line 159), “q-dependent).Python”, p. 7, “[38].The”, in Fig. 1, “E,mEv”, “625K”, “q,a*”.
- in some places in the text, the dot should be added, e.g. p. 1 (line 14), at the end of Abstract, (line 21) “[5] At”, p. 7 (line 176).
- in some places in the text, the dot should be deleted, e.g. in the paper's title.
- on p. 3 (lines 115, 116) and in the title of Fig. 1 is the same sentence: “q- is the deviation from the M-point as given above and E - energy transfer in meV.”
- on p. 2 (line 86) in “M3”, “3” should be given in superscript,
- on p. 3 (line 92), is “temperature”, but should be “Temperature”,
- on p. 7 (line 182), is “table 1”, but should be “Table 1”.
- References should be unified according to Instructions for Authors, e.g. full names of the title of journals are given, but the name of the journal title should be given in abbreviation. In Ref. [41] is et al., but all co-authors should be given. Pages (from-to) of all cited papers should be given. In Ref. [40] is “A.t”, but should be “A.T.”.
- English should be carefully checked, e.g.
- on p. 1 (line 17), is “antiferroelctric”, but should be “antiferroelectric”,
- in the manuscript, is e.g. “the Figure 1, the Figure 2, the Figure 3a, the Figure 3b, the equation 3, the modes 1 and 2, the section 3.1”, “the matrix 2”, but should be “Figure 1, Figure 2, Figure 3a, Figure 3b, equation 3, modes 1 and 2, section 3.1”, “matrix 2”.
- on p. 2, (lines 39), “as shown in the paper [14]”, (line 67), “In the very recent paper [25]” (line 201) (see Figure 2) and the interference term (see Figure 2) we can see” sound not good.
According to mentioned above remarks I suggest that in this paper the minor revision is needed before publication in Materials.
Reviewer 2 Report
This paper reports a study of lattice dynamics in PZT2.4 with inelastic X-ray scattering performed at different temperatures in the vicinity of the M point. The papers goes into a detailed analysis of the mode coupling to explain the shape of the IXS spectrum. Overall, the analyses are clearly presented in details and allow the reader to make an informed opinion. The proposed scenario makes sense.
I have essentially one remark: in the treatment of such IXS data, it is sometimes necessary to include a central line for some quasi-elastic scattering. Here the authors do not do so, and unless I am mistaken, this is not commented on. What is the reason that allows to ignore any quasi-elastic contribution? I am concerned that this might change the outcome of the fits, given that everything is very much concentrated in the low energy region.
I would also suggest some improvement in the figures, in that the authors might want to include more information (Q point for figure 2, temperature for figure 3) directly on the plots themselves rather than relying on the figure caption to do so, the reader would be grateful.
I recommend publication of the paper after these comments have been taken into account.
